# Evaluation of Success of Superhydrophobic Coatings in the Oil and Gas Construction Industry Using Structural Equation Modeling

**Ahsan Waqar** [1,*] , **Idris Othman** [1] , **Krzysztof Skrzypkowski** [2] and **Ali Shaan Manzoor Ghumman** [3,4]

1. Department of Civil & Environmental Engineering, University Technology PETRONAS, Seri Iskandar 32610, Perak, Malaysia
2. Faculty of Civil Engineering and Resource Management, AGH University of Science and Technology, Mickiewicza 30 Av., 30-059 Krakow, Poland
3. Chemical Engineering Department, Universiti Teknologi PETRONAS, Seri Iskandar 32610, Perak, Malaysia
4. HICoE, Centre for Biofuel and Biochemical Research (CBBR), Institute of Self-Sustainable Building, Universiti Teknologi PETRONAS, Seri Iskandar 32610, Perak, Malaysia
* Correspondence: ahsan_21002791@utp.edu.my

**Abstract:** In the oil and gas construction industry, the adoption of superhydrophobic coatings is still in the early adoption phase. Due to the lack of research and the importance of hydrophobic coatings in the oil and gas construction business, this study examined the success determinants of superhydrophobic coatings in Malaysia. This quantitative study included a pilot survey to assess questionnaire validity and Exploratory Factor Analysis (EFA) to reduce success variables discovered through a literature review. A structural equation modeling (SEM) approach was used to develop a model involving success factors of superhydrophobic coatings in the oil and gas construction industry of Malaysia. Four constructs in total were found in SEM, namely, performance success, sustainability construct, oil spill management, and safety and economic success. In total, five items were excluded from the model because their loading factors were less than 0.6. All Cronbach Alpha reliability constants were greater than 0.7, the composite reliability indicators were greater than 0.8, and the AVE was greater than 0.6 for all of the constructs, confirming acceptable reliability and validity statistics. Both convergent and discriminant validity confirmed the relationships between all constructs and the latent variable. The observed path coefficients between the constructs and the latent variable were 0.476 for performance success, 0.461 for sustainability success, 0.322 for oil spill management, and 0.242 for safety and economic success. The significance value for all of the constructs was less than 0.05, confirming the strong relationship between the constructs and the critical success of superhydrophobic coatings in the oil and gas industry.

**Keywords:** success; superhydrophobic coatings; oil and gas; construction industry; structural equation modeling

## 1. Introduction

In recent years, superhydrophobic coatings have attracted considerable interest owing to their potential to deliver several advantages to the oil and gas construction sector. These coatings are renowned for their ability to withstand liquids, especially oil, and provide better protection against extreme environments such as elevated temperatures, corrosive agents, and ultraviolet radiation [1,2]. In accordance with Ahmad et al. (2023), the exceptional qualities of superhydrophobic coatings originate from a confluence of chemical and physical causes, such as the presence of materials with low surface energy and the creation of hierarchical layer structures of high roughness and slight contact angles [3].

A previous study revealed that superhydrophobic coatings in the oil and gas sector have the potential to introduce several advantages, including enhanced equipment operation, less downtime, higher energy efficiency, and enhanced sustainability [4]. In addition,

by lowering the ecological impact of oil spillage, superhydrophobic coatings may assist in boosting the sustainability of oil and gas operations and preserve the environment from oil's destructive impacts [5].

Ahmad et al. (2023) argued that, despite the potential advantages of using superhydrophobic coatings, their performance in the oil and gas construction sector is still a subject of continuous study [3]. Several variables may affect the effectiveness of such coatings, including the oil used, the operating circumstances, and the type of coated equipment [6]. Complex and difficult to quantify is the effect of superhydrophobic coatings on the overall performance of the oil and gas sector [7,8]. To better appreciate the performance of these coatings, many elements must be considered, including the coatings' longevity, the effectiveness of oil processing and transportation systems, the sustainable site planning and management of oil spills, and the safety of activities [9,10]. A rigorous and systematic methodology that considers the interrelations between these aspects is required to assess the performance of superhydrophobic coatings within the oil and gas construction sector [3].

Structural Equation Modeling (SEM) is a statistical method often used to analyze the connections between several variables and to test hypotheses regarding how these variables impact one another [11]. SEM enables researchers to evaluate the direct and indirect impacts of many variables on the success of superhydrophobic coatings in the oil and gas construction sector and identify the most essential aspects that influence their success [12].

This research aims to assess the efficacy and success of superhydrophobic coatings inside the oil and gas construction sector using SEM [13,14]. The results of this investigation will provide insight into the viability of such coatings, and it will influence future research and development in this field [3]. The application of superhydrophobic coatings in the oil and gas sector is a relatively young and growing subject, and there is still much to learn about their performance in this industry [15,16]. This research is unusual in that it evaluates the performance of superhydrophobic coatings inside the oil and gas construction sector utilizing a cutting-edge, systematic methodology using SEM. This research will provide fresh and useful insights into the performance of superhydrophobic coatings in the oil and gas construction sector and aid in creating new and more effective coatings that are better suited to the industry's unique demands. The findings of this study will be of considerable interest to academics, engineers, industry professionals, politicians, and regulators who are attempting to encourage sustainable and responsible practices in the oil and gas business. This study's findings will also be useful for politicians and regulators trying to encourage sustainable and responsible behavior in the oil and gas sector. By gaining a deeper knowledge of the performance of superhydrophobic coatings throughout the oil and gas construction sector, policymakers and regulators can support the usage of these coatings and promote their widespread acceptance throughout the industry.

## 2. Related Studies

Numerous research studies have explored the effectiveness, durability, and potential environmental impact of superhydrophobic coatings in the oil and gas sector, which is a rapidly expanding subject. The effectiveness of superhydrophobic coatings considering the oil and gas construction sector remains largely unknown, despite the expanding body of research in this field [17,18]. For instance, Lei et al. (2020) discovered that superhydrophobic coatings decreased the spreading of oil by up to 95% compared to uncoated surfaces [19]. Yanlong et al. (2016) observed in a separate investigation that superhydrophobic coatings considerably decreased the quantity of oil attached to surfaces and avoided the development of oil slicks [20].

Additionally, the endurance and lifetime of superhydrophobic coatings have been investigated in prior studies. Multiple investigations have shown that these coatings are resilient and can endure exposure to unfavorable environmental conditions, including salt water, hot temperatures, and toxic materials [21,22]. For instance, Liang et al. (2020) discov-

ered that superhydrophobic coatings have been very resistant to salt water and chemical attack and remained effective even after constant exposure to these circumstances [23]. Lei et al. (2019) discovered in a separate investigation that superhydrophobic coatings maintained efficiency in preventing oil leaks after exposure to extreme temperatures and severe UV radiation [24]. In addition to its effectiveness and longevity, superhydrophobic coatings' environmental impact has been the topic of several previous studies [25]. For instance, Song et al. (2019) discovered that superhydrophobic coatings lowered the cost of cleaning up an oil spill by up to 80% compared to uncoated surfaces [26]. Similarly, Nookala and Asmatulu (2016) discovered that superhydrophobic coatings considerably decreased the indirect costs related to oil spills, such as productivity losses and reputational harm [27]. An et al. (2020), who explored both wettability and hydrophobicity using superhydrophobic coatings, is one such example [28].

Zhang et al. (2021) studied the mechanical characteristics of superhydrophobic coatings and discovered that they have superior mechanical strength and abrasion resistance, making them appropriate for usage in difficult situations [29]. The mechanical characteristics of superhydrophobic coatings have also been discovered to strongly depend on the material employed. Ren et al. (2021) examined the self-cleaning capabilities of superhydrophobic coatings and discovered that they are very efficient in removing dirt, oil, and other surface pollutants [30]. Rezaei et al. (2019) examined the electrical conductivity of superhydrophobic coatings and discovered that they were very conducive and their electrical conductivity greatly depended on the material utilized in their production [31]. Li et al. (2021) evaluated the impact of superhydrophobic coatings on the drag coefficient of ships and discovered that they greatly lowered the coefficient, hence increasing the fuel economy of ships [32].

Another research study by Bakhtiari et al. (2021) studied the impact of superhydrophobic coatings on adherence to biofouling [33]. The research discovered that superhydrophobic coatings may effectively inhibit biofouling development and enhance surface antifouling performance. Talens-Perales et al. (2021) examined the corrosion prevention capability of superhydrophobic coatings [34]. The research discovered that superhydrophobic coatings have successfully protected surfaces against corrosion and might increase the durability of materials used in severe conditions.

Kiran Raj (2021) examined the durability and lifetime of superhydrophobic coatings under different environmental conditions [35]. The research discovered that superhydrophobic coatings may retain their hydrophobic qualities for extended periods, even under extreme environmental circumstances, making them appropriate for application in oil and gas construction.

Tarannum et al. (2020) investigated the influence of superhydrophobic coatings on the evaporation of water. The research discovered that superhydrophobic coatings may greatly lower the rate of water evaporation, making them appropriate for use in industries where water conservation is crucial, such as the oil and gas construction sector [36]. These research findings provide vital insight into the performance of superhydrophobic coatings that emphasize their potential for industry-wide advancement [37,38]. According to Sarubbo et al. (2022), the durability of superhydrophobic coatings is one of its key disadvantages [6]. Due to exposure to adverse environmental factors such as high temperatures, UV rays, and mechanical wear, these coatings might deteriorate over time and become less efficient at shielding the underlying structures. The price of superhydrophobic coatings is another significant disadvantage. Many oil and gas building projects may not be able to afford these coatings in the long run due to their high cost. According to Peng et al. (2021), superhydrophobic coatings might not work well with some of the metals, polymers, and composites utilized in the oil and gas construction sector [39]. This might reduce their ability to prevent corrosion in equipment and structures. Superhydrophobic coating application can be a challenging and time-consuming operation that calls for specific expertise and tools. For the oil and gas construction sector, this may mean higher prices and less efficiency. Superhydrophobic coatings might not be successful at repelling all liquids and materials,

including grease and oil, which are frequently found in the oil and gas construction sector. This may reduce how well these coatings protect equipment and structures from harm caused by these contaminants. Table 1 is presenting all success factors identified from literature for superhydrophobic coatings application in oil and gas sector.

Unique to this research is the use of structural equation modeling (SEM) to assess the effectiveness of superhydrophobic coatings in the oil and gas construction sector. SEM is a strong statistical instrument that enables the investigation of complicated connections between variables and may offer a deeper understanding of the performance of superhydrophobic coatings [39,40]. Prior research has generally focused on the specific advantages of superhydrophobic coatings, such as their ability to reduce oil spread and prevent biofouling. This research examines the industry-wide success of superhydrophobic coatings and the linked elements that contribute to that achievement. In addition, by using SEM, this research can evaluate the intensity and direction of the correlations between variables, offering greater knowledge of the performance of superhydrophobic coatings and the contributing elements. This may provide important data for further research and the enhancement of superhydrophobic coatings inside the oil and gas construction sector.

**Table 1.** Identified success factors for superhydrophobic coating application in oil and gas sector.

| Code | Success Factors | References |
|---|---|---|
| S1 | Superhydrophobic coatings may enhance the effectiveness of oil processing & transportation systems, resulting in energy savings. | [2,41] |
| S2 | The resistance of superhydrophobic coatings to adverse states of the environment, including high temperatures, humidity levels, and chemical exposure, makes them excellent for application in the oil and gas sector. | [4,42] |
| S3 | Superhydrophobic coatings may dramatically increase a surface's capacity to resist liquids, including oil, making them particularly useful for managing oil spills. | [21,22] |
| S4 | Companies in the oil and gas sector that utilize superhydrophobic coatings may gain a competitive edge by demonstrating their dedication to environmental preservation and operational efficiency. | [23,24] |
| S5 | Superhydrophobic coatings may enhance the operation and performance of oil and gas sector equipment. | [27,43] |
| S6 | These coatings may improve the effectiveness of oil processing as well as transport networks by lowering the quantity of oil that clings to surfaces. | [28,29] |
| S7 | By minimizing the likelihood of fires and other risks connected with oil spillage, superhydrophobic coatings may enhance the safety of the oil and gas industry's personnel in the workplace. | [30,31] |
| S8 | By preventing oil from clinging to objects and preventing corrosion, superhydrophobic coatings may extend the life of oil and gas sector equipment. | [44,45] |
| S9 | By minimizing the requirement for maintenance and upkeep, superhydrophobic coatings may decrease the needed downtime for oil and gas sector equipment. | [32,33] |
| S10 | By shielding equipment from the corrosive effects of oil as well as other fluids, superhydrophobic coatings may enhance the resilience of oil and gas sector equipment. | [35,37] |

**Table 1.** *Cont.*

| Code | Success Factors | References |
|---|---|---|
| S11 | By lowering the environmental effect of oil leaks, superhydrophobic coatings may aid in enhancing the sustainability of oil and gas sector operations. | [36,38] |
| S12 | Superhydrophobic coatings may help businesses comply with environmental requirements and standards by lowering the danger of oil spills as well as the environmental effect of spills. | [46,47] |
| S13 | Superhydrophobic coatings may minimize the quantity of oil that enters the environment after oil spills, hence minimizing the harm to species and ecosystems. | [48,49] |
| S14 | By decreasing the time necessary to clean up oil spills and the requirement of maintenance, superhydrophobic coatings may aid in increasing manufacturing productivity. | [50,51] |
| S15 | Superhydrophobic coatings may make surfaces simpler to clean, saving the time and effort required to keep equipment and infrastructure. | [52,53] |
| S16 | Companies that utilize superhydrophobic coatings to prevent oil spills may enhance their image by showing their dedication to eco-friendly preservation and operational efficiency. | [46] |
| S17 | The continuous usage and effectiveness using superhydrophobic coatings within the oil and gas sector may motivate additional development and research of these materials, resulting in the future fabrication of coatings that are even more efficient and effective. | [47,54] |
| S18 | By lowering the need for regular cleaning, superhydrophobic coatings may assist firms in the oil and gas sector reduce their maintenance costs. | [55,56] |
| S19 | The application of superhydrophobic coatings may lower the expense of clearing up oil spills and repairing damaged infrastructure and equipment. | [57,58] |
| S20 | Superhydrophobic coatings may increase the performance of substances used in the oil and gas sector by enhancing their resistance to oil and other liquids. | [59,60] |
| S21 | In addition to repelling fluids, superhydrophobic coatings may boost the slip resistance on surfaces, hence enhancing safety and decreasing the chance of accidents. | [61,62] |
| S22 | Superhydrophobic coatings may help businesses comply with conservational requirements and standards by lowering the danger of oil spills as well as the environmental effect of spills. | [63] |
| S23 | Superhydrophobic coatings may boost the dependability of oil and gas sector equipment by avoiding oil from adherent to objects and causing corrosion. | [64,65] |
| S24 | Superhydrophobic coatings may make it simpler and quicker to clear up oil spills, hence reducing the time needed to resume regular operations. | [66,67] |
| S25 | By preventing oil from clinging to the surface, superhydrophobic coatings may lessen the risk of fire and other dangers related to oil spills. | [68,69] |

This work provides a new technique to assess the performance of superhydrophobic coatings and the potential to contribute to a deeper knowledge of their influence on the industry. This may assist in advancing the development of superhydrophobic coatings and enhance their performance in real-world applications.

## 3. Methodology

Using structural equation modeling, the current research intends to assess the performance of superhydrophobic coatings throughout the oil and gas construction business. The theoretical model was developed by synthesizing a complete literature study, which resulted in the formation of transitional ideas (or hypotheses) confirmed by experimental evidence. The conceptual modeling approach consisted of three steps: (i) Finding the

model's components, (ii) organizing the model's components, and (iii) establishing the model's components' linkages. Figure 1 displays the research methodology.

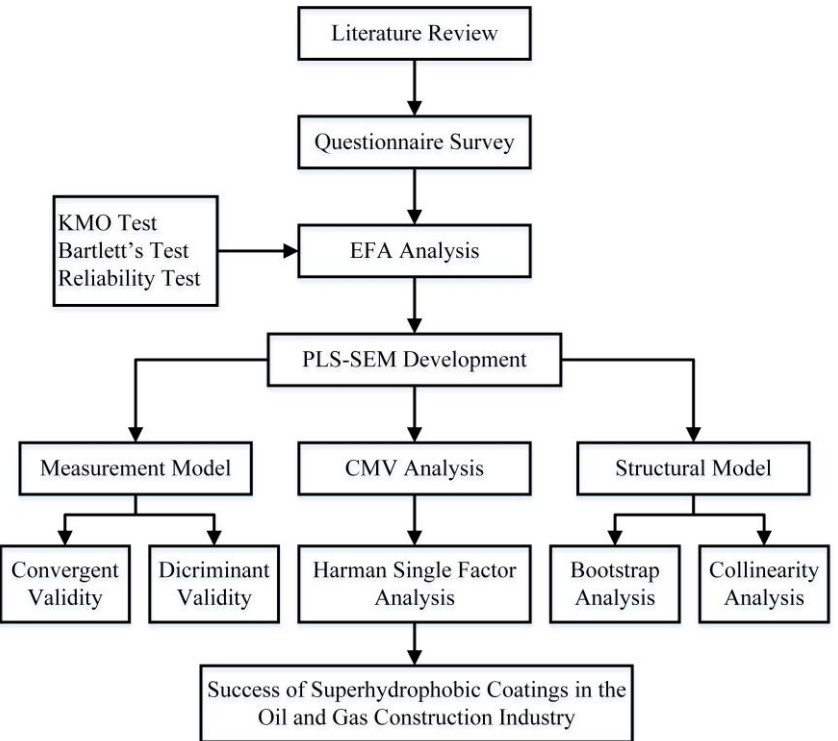

**Figure 1.** Research methodology flowchart.

Given the relatively recent introduction of superhydrophobic coating in Malaysia, the research design was based on the findings of Nikolova and Gutierrez (2021) and Sarubbo et al. (2022), and a stratified sample method was used to reach a particular subpopulation [6,42]. According to Xu et al. (2021), the benefits of stratified sampling include the reduction of bias in sample case selection and the ability to generalize to the population, considering distinctions by stratification and encompassing all three areas (client, contractor, and consultant) in Malaysia [70].

Using a five-point Likert scale, respondents rated their experiences and understanding of superhydrophobic adoption advantages, with scores of 5 and 1 indicating extremely high and very low, respectively, and scores of high, medium, and low lying in between. The sample size was chosen by reviewing the relevant literature, and more than thirty instances were deemed sufficient for future investigation [2,41]. Multiple sources were used to establish the research sample size, and more than 30 instances were deemed suitable for analysis. In contrast to previous research, which needs a sample size of 200 or more for a comprehensive SEM analysis, this study recruited 144 participants out of 230 construction professionals, with a 67% response rate. This sample size was deemed enough for SEM analysis [5,42].

This research makes a significant addition to the area by providing a systematic and rigorous assessment of the performance of superhydrophobic coatings throughout the oil and gas construction sector [9,70]. These research findings might provide construction sector participants with realistic solutions based on the methods used in various oil and gas projects.

The research was undertaken in Malaysia, where superhydrophobic coatings are still relatively novel [2,5]. For research on superhydrophobic coating application, a stratified sampling strategy was used to obtain data from a specific subpopulation group since it is thought to produce the most reliable and accurate results.



### 3.1. Exploratory Factor Analysis

Exploratory Factor Analysis (EFA) would be a frequently adopted statistical method for identifying and examining the underlying structure of a collection of variables. It helps determine if there is a link among the variables and how they are connected. In the present research framework, EFA was performed to evaluate the Malaysian oil and gas sector groups using a survey issued to field specialists [4,41].

Typically, the sample size of EFA is between 150 and 300 observations [71]. Nevertheless, other researchers suggest that investigators have some leeway concerning sample size, and selecting sample sizes that are proportional to the number of variables involved is advised. It is estimated that the optimal number of variables needed for factor analysis is between 20 and 50 [1,72]. Whenever the number of variables reaches this limit, individual features cannot be identified satisfactorily, and certain studies may need fewer variables if the sample size is large enough [35,36].

In this research, the 25 discovered success variables and the complete questionnaires collected from 144 respondents were judged appropriate for EFA. The population used in the current research was deemed a representative sample over all relevant ranges. EFA is a key stage in data analysis since it simplifies and reduces the number of variables, making it simpler to comprehend their correlations [37,38]. In addition, EFA assists in identifying patterns in the information and revealing any hidden underlying structure. The findings of the EFA will provide useful insights into the performance of superhydrophobic coatings throughout the oil and gas construction sector and serve as a platform for future statistical analysis in the present research.

### 3.2. Analytical Model

To better comprehend the use of superhydrophobic coatings throughout the oil and gas sector, it is necessary to examine the numerous models that have been utilized to investigate this topic. While there are several models that have been utilized previously, it is essential to choose the one that best matches the objectives of the study [40]. Multiple linear regression (MLR), structural equation modelling (SEM), artificial neural networks (ANNs), and system dynamics (SD) are the most often used models. Multiple linear regression (MLR) would be a statistical method for predicting a dependent variable using several independent variables. Although this model has been used extensively in several research studies, it is not necessarily the ideal option for analyzing the effectiveness of superhydrophobic coatings [39,68]. This is because MLR does not include the correlations among non-observed variables, which might be a significant limitation of this model.

Structural equation modelling (SEM) is another prevalent approach that may be used to examine the effectiveness of superhydrophobic coatings. This model is very valuable since it can explain the connection between several observable and unobservable variables. Utilized widely in the construction sector, SEM has been shown to be a significant technique for identifying faults within variables [66,69]. In this research, the partial least squares (PLS) model was used to determine the link between the application and success of superhydrophobic coatings in the oil and gas industry [54,56]. The PLS-SEM model was used to evaluate the model and verify its resemblance to the data, in addition to testing the parameter assumptions [46,47].

In this study, one of the objectives of the data reduction technique was to decrease the number of parameters and variables in the research model to a tolerable level relative to the size of the sample determined by the SEM ratio. This was performed to guarantee that the model was as accurate as possible while also minimizing the possibility of data inaccuracies [51,52].

Convergent validity is the degree to which several measurements of a concept or variable give comparable findings, and hence converge to suggest the identical underlying construct. In the context of research analyzing the performance of superhydrophobic coatings throughout the oil and gas construction sector using structural equation modelling, it is essential to examine convergent validity to assure the accuracy and dependability of

success metrics [50,53]. When assessing the validity of a measuring instrument or research model, it is essential to include discriminant validity. In the context of employing structural equation modelling to evaluate the performance of superhydrophobic coatings throughout the oil and gas construction sector, discriminant validity refers to the model's capacity to differentiate between the variables it is supposed to assess [49,50].

This implies that the model's variables should not be strongly linked or loaded on the same factor, since this would indicate that they measure the same concept. To evaluate discriminant validity, researchers may use a variety of techniques, such as measuring the component loadings and analyzing the correlations between variables. In structural equation modelling, this may be accomplished by comparing the magnitude of the loadings and analyzing the pattern of the variable relationships [66,68]. If somehow the factor loadings for a specific variable are relatively large and the correlations across variables are minimal, this indicates that the model has high discriminant validity. Comparing the variable factor loadings and correlations with a set of expectations or standards is another method for determining discriminant validity [67,69]. For instance, if two variables are predicted to have a strong association, but their loadings and correlation are low, this may indicate that the model has poor discriminant validity.

In conclusion, while analyzing the effectiveness of superhydrophobic coatings, it is crucial to use the appropriate model. SEM was selected for this study as it has been extensively used in studies on the construction sector and has been found to be an efficient method for testing hypothesized associations concurrently [37,40]. The partial least squares (PLS) model was used to determine the correlations between the application and success of superhydrophobic coatings, while the PLS-SEM model was utilized to verify the model and its assumptions [9,71]. Researchers may acquire greater knowledge of the effectiveness of superhydrophobic coatings and their influence on the oil and gas sector with the proper model.

## 4. Results

### 4.1. Respondent Characterization

The features of the respondents in the research superhydrophobic coatings in the oil and gas construction sector using structural equation modelling were evaluated in order to comprehend the participants' demographics and evaluate the sample's representativeness. A self-administered questionnaire was issued to specialists in the oil and gas sector to gather the data.

The percentages shown here represent the following qualities of the respondents:

- The majority of responders were between the ages of 41 and 50 (35.5%), followed by those between 31 and 40 (29.5%) and 51 to 60 (22.1%). The remaining individuals were either younger than 30 (6.4%) or older than 60 (6.8%).
- Gender: 88.0% of the sample comprised male individuals, whereas 12.0% comprised female participants.
- Most respondents (52.2%) had a master's degree, followed by those with a bachelor's (39.5%). The remaining individuals either had a Ph.D. (6.2%) or even a high school education (2.2%).
- The bulk of respondents had 20 years of oil and gas sector experience (54.4%), followed by 11–20 years (29.4%) and 1–10 years (16.3%).
- A considerable percentage of the respondents held a managerial role (43.3%), accompanied by a technical role (35.5%) and an administrative position (21.3%).
- Many participants (46.5%) worked for firms with more than 500 workers, followed by those with between 51 and 500 employees (40.4%), and those with fewer than 50 staff (13.1%).
- Most of the enterprises in the sample were private (76.2%), followed by state-owned (17.3%) and cooperative (6.5%) businesses.

These features of the respondents provide useful data on the sample and facilitate comprehension of the data's representativeness. Participants in the research came from a

variety of professional experiences and backgrounds, which strengthens the generalization of the study. The study considered the participants' age, gender, education level, job experience, position in the firm, company size, and company type to ensure that the findings were not skewed toward any one demographic group.

*4.2. Exploratory Factor Analysis*

Using EFA, we looked at 25 items pertaining to the advantages of using superhydrophobic coating applications in the oil and gas sector for the building of the model, and several well-known factorability parameters were used. Commonly used to determine whether partial connections between items are insignificant, the KMO is a uniformity of factor dimension [7,8]. For reliable factor analysis, the KMO index must be between 0 and 1, and 0.6 is the minimum acceptable value [14]. Whether or not the correlation matrix is the same may also be determined using Bartlett's sphericity test. To conduct a valid component analysis, Reis et al. (2022) suggest using the sphericity test developed by Bartlett. When the probability value is less than 0.05, the result is deemed significant [16].

Kaiser–Meyer–Olkin sample appropriateness ratios of 0.830 were found for using superhydrophobic coating applications in the oil and gas sector. These numbers are larger than what is considered acceptable for the sphericity test according to Bartlett's scale. Using superhydrophobic coating application in the oil and gas sector ($x2$ (144) = 470.008, $p < 0.05$) benefited significantly. The anti-correlation image's matrix has putative values of over 0.5 throughout its entire diagonal after controlling for individual components using factor analysis. Lower values (0.3) for the preliminary communalities denote variables that do not suit the solution factor, while higher values indicate that the variation within the variables is greater than the variance between all components. All initial communalities in this study were higher than the cutoff, and all factor loadings were positive. The results of EFA analysis are presented in Table 2.

**Table 2.** EFA results.

| Variables | Component | | | | Cronbach Alpha |
|---|---|---|---|---|---|
| | **1** | **2** | **3** | **4** | |
| S1 | 0.818 | | | | 0.874 |
| S24 | 0.774 | | | | |
| S14 | 0.758 | | | | |
| S8 | 0.718 | | | | |
| S9 | 0.696 | | | | |
| S21 | 0.624 | | | | |
| S10 | 0.510 | | | | |
| S2 | | 0.789 | | | 0.861 |
| S25 | | 0.744 | | | |
| S13 | | 0.737 | | | |
| S11 | | 0.581 | | | |
| S7 | | 0.558 | | | |
| S4 | | 0.532 | | | |
| S22 | | | 0.640 | | 0.787 |
| S23 | | | 0.637 | | |
| S3 | | | 0.629 | | |
| S6 | | | | 0.740 | 0.776 |
| S19 | | | | 0.648 | |
| S15 | | | | 0.563 | |
| Eigen Value | 4.775 | 3.270 | 2.469 | 2.362 | |
| % Variance | 19.099 | 13.082 | 9.874 | 9.448 | |

Excluded Variables: S5, S12, S18, S20, S17, and S16 are extracted due to cross-loading or loading less than 0.5.

Four factors with eigenvalues of 1 or greater were found in exploratory factor analysis of all 19 aspects related to superhydrophobic coating application. Four components adequately explained 51.503% of the total variance, as measured by eigenvalues. As can be seen in Table 2, S5, S12, S18, S20, S17, and S16 were extracted due to cross-loading or loading less than 0.5 and were left out of the main analysis because of cross-loading. There were four components found to be significant by EFA, all of which had an impact on the robotics derivers, and three of these factors had eigenvalues greater than 1.

The statistical validity of the components derived via EFA was evaluated. The greatest factor loading for every parameter in the matrix organizational structure was used to calculate factor scores for the various stages of the factor (or group). Table 3 is presenting the named construct along with their specific factors identified from EFA. According to the data table, the dependability evaluation also passed with acceptable results, indicated in Table 4. Although the average value was 0.7 and values more than 0.8 were considered credible, Sebastian et al. (2021) state that the Cronbach alpha value for newly generated dimensions should be larger than 0.6 [7]. Cronbach's alpha was more than 0.6, and the average correlation between variables was greater than 0.3 across all components, indicating that there was harmony among the internal variables.

**Table 3.** Grouped success factors based on EFA findings.

| Group | Assigned Code | Success Factors |
|---|---|---|
| Performance Success | S1 | Superhydrophobic coatings may enhance the effectiveness of oil processing & transportation systems, resulting in energy savings. |
| | S24 | Superhydrophobic coatings may make it simpler and quicker to clear up oil spills, hence reducing the time needed to resume regular operations. |
| | S14 | By decreasing the time necessary to clean up oil spills and the requirement of maintenance, superhydrophobic coatings may aid in increasing manufacturing productivity. |
| | S8 | By preventing oil from clinging to objects and preventing corrosion, superhydrophobic coatings may extend the life of oil and gas sector equipment. |
| | S9 | By minimizing the requirement for maintenance and upkeep, superhydrophobic coatings may decrease the needed downtime for oil and gas sector equipment. |
| | S21 | In addition to repelling fluids, superhydrophobic coatings may boost the slip resistance on surfaces, hence enhancing safety and decreasing the chance of accidents. |
| | S10 | By shielding equipment from the corrosive effects of oil as well as other fluids, superhydrophobic coatings may enhance the resilience of oil and gas sector equipment. |
| Sustainability Success | S2 | The resistance of superhydrophobic coatings to adverse states of the environment, including high temperatures, humidity levels, and chemical exposure, makes them excellent for application in the oil and gas sector. |
| | S25 | By preventing oil from clinging to the surface, superhydrophobic coatings may lessen the risk of fire and other dangers related to oil spills. |
| | S13 | Superhydrophobic coatings may minimize the quantity of oil that enters the environment after oil spills, hence minimizing the harm to species and ecosystems. |
| | S11 | By lowering the environmental effect of oil leaks, superhydrophobic coatings may aid in enhancing the sustainability of oil and gas sector operations. |
| | S7 | By minimizing the likelihood of fires and other risks connected with oil spillage, superhydrophobic coatings may enhance the safety of the oil and gas industry's personnel in the workplace. |
| | S4 | Companies in the oil and gas sector that utilize superhydrophobic coatings may gain a competitive edge by demonstrating their dedication to environmental preservation and operational efficiency. |

**Table 3.** *Cont.*

| Group | Assigned Code | Success Factors |
|---|---|---|
| Oil Spill Management | S22 | Superhydrophobic coatings may help businesses comply with conservational requirements and standards by lowering the danger of oil spills as well as the environmental effect of spills. |
| | S23 | Superhydrophobic coatings may boost the dependability of oil and gas sector equipment by avoiding oil from adherent to objects and causing corrosion. |
| | S3 | Superhydrophobic coatings may dramatically increase a surface's capacity to resist liquids, including oil, making them particularly useful for managing oil spills. |
| Safety and Economic Success | S6 | These coatings may improve the effectiveness of oil processing as well as transport networks by lowering the quantity of oil that clings to surfaces. |
| | S19 | The application of superhydrophobic coatings may lower the expense of clearing up oil spills and repairing damaged infrastructure and equipment. |
| | S15 | Superhydrophobic coatings may make surfaces simpler to clean, saving the time and effort required to keep equipment and infrastructure. |

**Table 4.** Model reliability statistics.

| Group | Assigned Code | Initial Loadings | Final Loadings | Cronbach Alpha | Composite Reliability | AVE | SE |
|---|---|---|---|---|---|---|---|
| Performance Success | S1 | 0.900 | 0.918 | 0.91 | 0.933 | 0.737 | 0.084 |
| | S24 | 0.883 | 0.901 | - | - | - | 0.085 |
| | S14 | 0.842 | 0.852 | - | - | - | 0.095 |
| | S8 | 0.785 | 0.806 | - | - | - | 0.083 |
| | S9 | 0.793 | 0.811 | - | - | - | 0.090 |
| | S21 | 0.563 | Deleted | - | - | - | - |
| | S10 | 0.389 | Deleted | - | - | - | - |
| Sustainability Success | S2 | 0.781 | 0.871 | 0.779 | 0.86 | 0.609 | 0.087 |
| | S25 | 0.763 | 0.852 | - | - | - | 0.090 |
| | S13 | 0.740 | 0.739 | - | - | - | 0.090 |
| | S11 | 0.616 | Deleted | - | - | - | - |
| | S7 | 0.662 | Deleted | - | - | - | - |
| | S4 | 0.691 | 0.638 | - | - | - | 0.092 |
| Oil Spill Management | S22 | 0.918 | 0.940 | 0.85 | 0.93 | 0.869 | 0.096 |
| | S23 | 0.897 | 0.924 | - | - | - | 0.087 |
| | S3 | 0.423 | Deleted | - | - | - | - |
| Safety and Economic Success | S6 | 0.827 | 0.811 | 0.700 | 0.827 | 0.616 | 0.093 |
| | S19 | 0.841 | 0.843 | - | - | - | 0.091 |
| | S15 | 0.671 | 0.693 | - | - | - | 0.086 |

*4.3. Common Method Bias*

Common method bias (CMB) arises when the same technique is used to assess dependent and independent variables in research, resulting in exaggerated associations between the variables. This is a possible problem in the context of the study because we examined the connections between various factors and the achievement of superhydrophobic coatings using survey data.

Harman's single component analysis, often known as the common method variance (CMV) approach, is one strategy to handle CMB. This technique entails finding a single underlying component that reflects the common way used to gather the data and then determining whether this variable accounts for a large amount of the data's variation. If this common technique element accounts for a substantial proportion of the variation, it shows that CMB may well be present, and the findings' validity may be compromised. This approach identifies if a single unifying component can account for a significant percentage of the data variation. If the number is below 50 percent, it indicates that common method bias is not a major problem.

### 4.4. Analytical Model

The analytical modeling was performed in order to perform a relationship analysis of each of the constructs involved and to determine the ultimate radical success of superhydrophobic coatings in the oil and gas industry. Following the structural equation modeling approach, the measurement model is constructed, after which, using the primary data, further validity analysis is conducted to determine the efficiency of the model in the context of future implications connected to the study. The different factors are considered in structural equation modeling but in accordance with the quantitative analysis, which is aimed at developing the final structural model involving all the success factors and their constructs.

#### 4.4.1. Convergent Validity

The statistical notion of convergent validity examines the agreement between different measurements of the same construct. Showing a strong correlation across measurements of a construct and proving that the construct is a good predictor of an outcome are common ways of establishing convergent validity in SEM. The reliability of a scale may be evaluated by calculating the internal consistency using a statistic called Cronbach's alpha. The findings show that the Cronbach alpha for all constructions is more than 0.7, indicating high levels of internal consistency. In SEM, the dependability of a composite (latent) construct is measured by its composite reliability. Based on the data, we can infer that the composite dependability is over 0.8 across the board for these constructions. Figures 2 and 3 indicates the variation in composite reliability and AVE, while Table 3 presents the overall findings from the convergent validity evaluation. Following the initial factor loadings, Figure 2 presents the SEM with insignificant factors while the complete model is presented in Figure 3 with only significant constructs and success factors. The AVE indicates how much of the total variation in a dependent variable can be accounted for by a given collection of independent variables. Trend of item leadings with AVE and Composite Reliability is presented in Figures 4 and 5. The AVE is used to determine the level of fit of a structural model in structural equation modeling. According to the data, AVE is more than 0.6 across the board, suggesting a satisfactory model fit. Most experts agree that a value of 0.5 or above for the AVE statistic indicates a satisfactory match.

#### 4.4.2. Discriminant Validity

In SEM, the Fornell–Larcker Criteria are used to assess the discriminant validity of a given measuring tool. According to the criterion, if the squared correlation between two constructs is lower than the AVE retrieved from each construct, then the two constructs cannot be deemed separate. Discriminant validity is thought to exist when the AVE is 0.5 or greater. Using the Fornell–Larcker criterion, if the AVE values are more than 0.5, then the findings may be considered valid, which is already evident from Table 4, while Table 5 further indicates acceptable results as the correlations between the constructs are acceptable. The convergent validity of a measurement model may be assessed using the HTMT criterion, which is a set of rules used in structural equation modeling. According to the criterion, a convergent construct has a higher square root of the AVE than the correlation with any other construct in the model. The common consensus is that AVE values of 0.5 or above imply strong convergent validity. In this scenario, HTMT levels over 0.7 are regarded as satisfactory as indicated in Table 6. Measurement model validity, as well as the discriminant and convergent validity of constructs, may be assessed with the use of the Fornell–Larcker criteria and the HTMT criterion, two key ideas in structural equation modeling. In both situations, AVE values in the 0.5–0.7 range are commonly accepted. This is true for the Fornell–Larcker criterion and the HTMT criterion. The cross-loading values are indicated in Table 7 where all values are higher than 0.6, indicating acceptable statistics for including the factors in the final SEM.

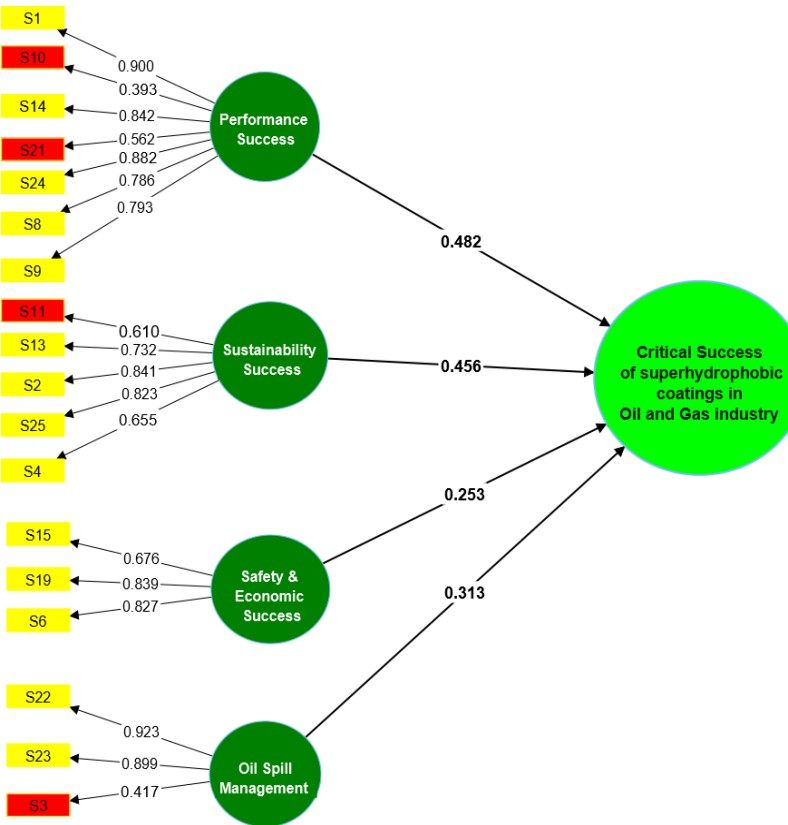

**Figure 2.** Initial model with insignificant factors.

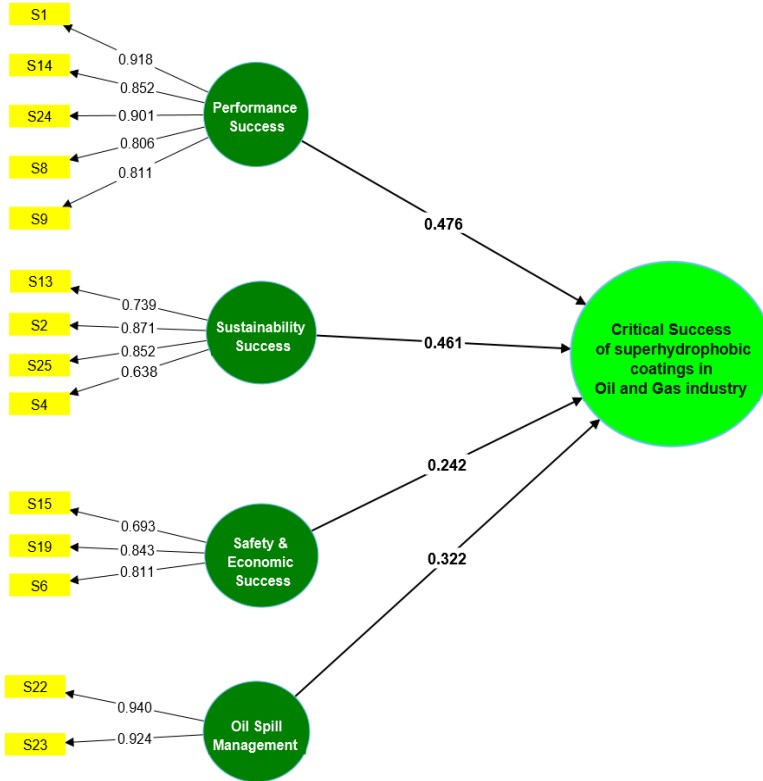

**Figure 3.** Optimized model with path coefficients.

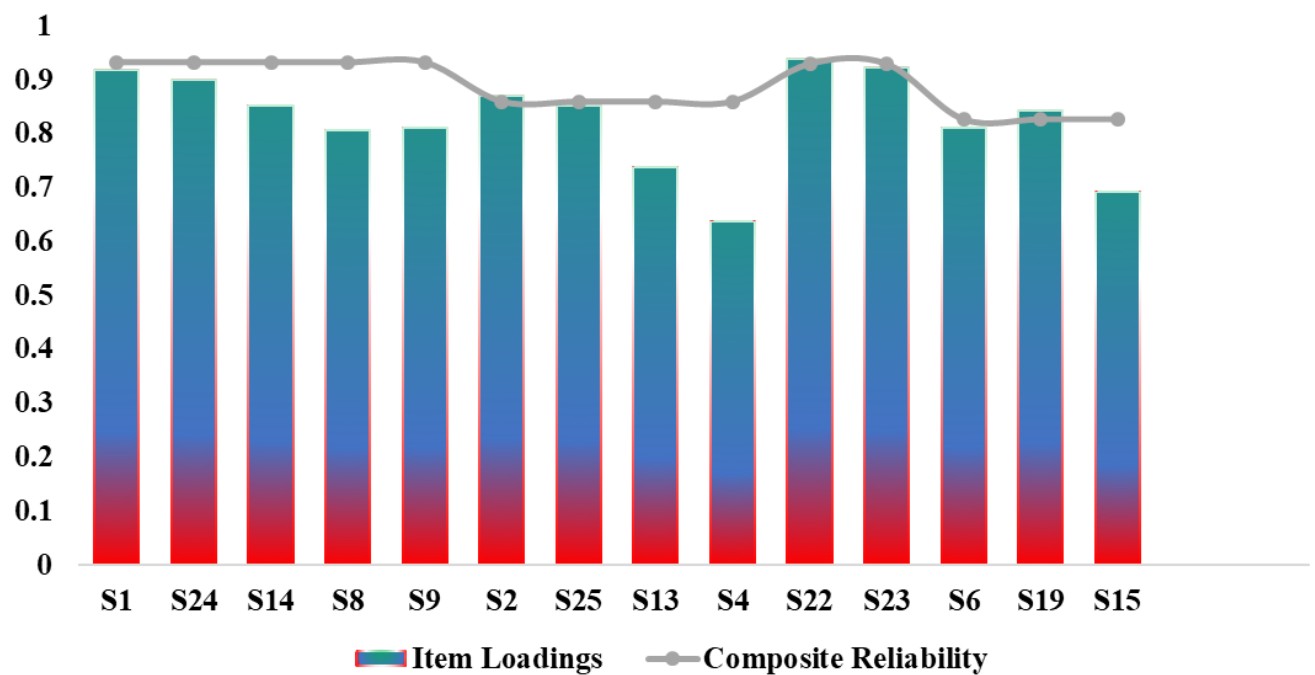

**Figure 4.** Item loadings vs. composite reliability chart.

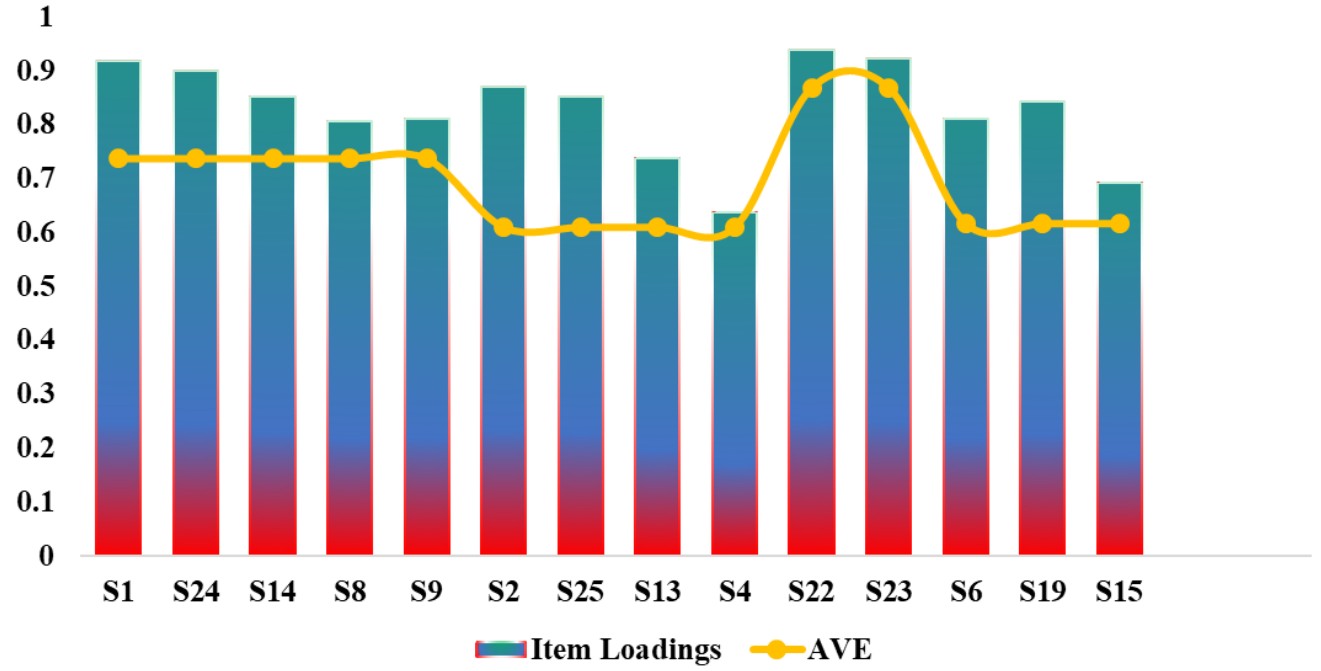

**Figure 5.** Item loadings vs. AVE chart.

**Table 5.** Fornell–Larcker statistics.

| Constructs | Oil Spill Management | Performance Success | Safety & Economic Success | Sustainability Success |
|---|---|---|---|---|
| Oil Spill Management | | | | |
| Performance Success | 0.219 | | | |
| Safety & Economic Success | 0.161 | 0.192 | | |
| Sustainability Success | 0.235 | 0.551 | 0.177 | |

**Table 6.** HTMT statistics.

| Constructs | Oil Spill Management | Performance Success | Safety & Economic Success | Sustainability Success |
|---|---|---|---|---|
| Oil Spill Management | 0.932 | | | |
| Performance Success | 0.195 | 0.859 | | |
| Safety & Economic Success | 0.133 | 0.156 | 0.785 | |
| Sustainability Success | 0.19 | 0.465 | 0.093 | 0.781 |

**Table 7.** Cross loadings observed from structural model analysis.

| Variables | Oil Spill Management | Performance Success | Safety & Economic Success | Sustainability Success |
|---|---|---|---|---|
| S22 | 0.94 | 0.198 | 0.123 | 0.213 |
| S23 | 0.924 | 0.163 | 0.126 | 0.137 |
| S1 | 0.171 | 0.918 | 0.195 | 0.422 |
| S24 | 0.18 | 0.901 | 0.141 | 0.453 |
| S14 | 0.218 | 0.852 | 0.152 | 0.395 |
| S8 | 0.106 | 0.806 | 0.061 | 0.341 |
| S9 | 0.154 | 0.811 | 0.107 | 0.375 |
| S6 | 0.107 | 0.072 | 0.811 | 0.053 |
| S15 | 0.051 | 0.15 | 0.693 | −0.081 |
| S19 | 0.136 | 0.149 | 0.843 | 0.174 |
| S2 | 0.109 | 0.329 | 0.086 | 0.871 |
| S4 | 0.195 | 0.323 | 0.063 | 0.638 |
| S25 | 0.116 | 0.355 | 0.092 | 0.852 |
| S13 | 0.178 | 0.438 | 0.047 | 0.739 |

### 4.4.3. Bootstrapping Structural Path Analysis

Performance success ($\beta = 0.476$, VIF = 1.313, SE = 0.03, t-stat = 8.142, $p = 0.000$), sustainability success ($\beta = 0.461$, VIF = 1.293, SE = 0.03, t-stat = 12.648, $p = 0.000$), safety and economic success ($\beta = 0.242$, VIF = 1.037, SE = 0.027, t-stat = 7.74, $p = 0.000$), and oil spill management ($\beta = 0.322$, VIF = 1.065, SE = 0.03, t-stat = 8.142, $p = 0.000$) constructs indicated a significant relationship with the latent variable, that is, the critical success of superhydrophobic coatings in oil and gas industry. Acceptable VIF values need to be less than 3.5, which indicates the validity of the path relationships of the variables involved in the model. $p$-values need to be less than 0.001, which eradicates the possibility of accepting null hypotheses for all the constructs as indicated in Table 8. SE coefficients need to be non-zero, which indicates the path validity of the model. The main path coefficient $\beta$ indicates the significance of the relationship of the constructs with the success of superhydrophobic coatings in the oil and gas industry. Figure 6 presents the model with all $p$-values indicating that the model can be used in future research where the success of superhydrophobic coatings in the oil and gas industry can be improved. The histograms presented in Figures 7–10 indicate the behavior of the data involved in analysis from a normal distribution perspective. Following the curve, the histograms indicate and confirm the validity of the model in terms of the significant impact of constructs in achieving the critical success of superhydrophobic coatings in the oil and gas industry. Depending on the results, it can be stated that the model is clearly validated. Furthermore, the normal distribution behavior is eradicated from the model evaluation results, which confirms the successful results.

**Table 8.** Path analysis results.

| Pathway | β | SE | *t*-Value | *p*-Value | VIF |
|---|---|---|---|---|---|
| Oil Spill Management -> Critical Success of Superhydrophobic Coatings in Oil and Gas industry | 0.322 | 0.03 | 8.142 | <0.001 | 1.065 |
| Performance Success -> Critical Success of Superhydrophobic Coatings in Oil and Gas industry | 0.476 | 0.03 | 20.046 | <0.001 | 1.313 |
| Safety & Economic Success -> Critical Success of Superhydrophobic Coatings in Oil and Gas industry | 0.242 | 0.027 | 7.74 | <0.001 | 1.037 |
| Sustainability Success -> Critical Success of Superhydrophobic Coatings in Oil and Gas industry | 0.461 | 0.03 | 12.648 | <0.001 | 1.293 |

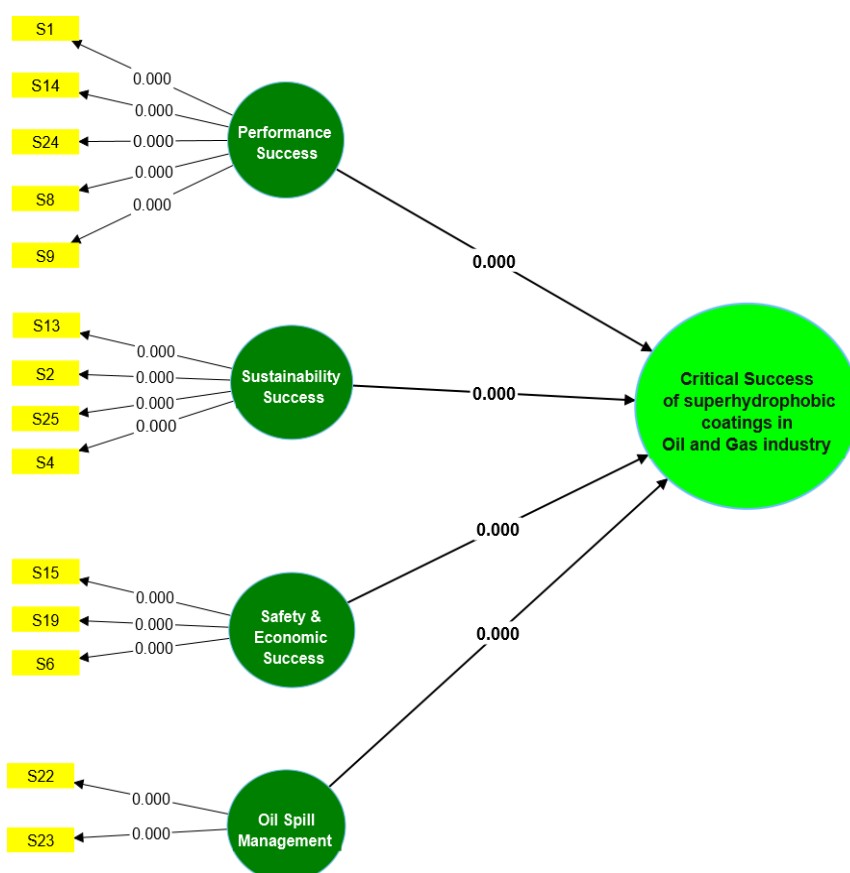

**Figure 6.** Model indicating path significance.

4.4.4. Predictive Relevance of the Structural Model

The System Sum of Squares (SSO) quantifies how much of the observed data's variability may be attributed to the model. A better model fit to data is indicated by a larger SSO value. The sum of squared residuals (SSE) quantifies the discrepancy between observed and expected values. The closer the SSE is to zero, the better the model matches the data. For quantifying how much of the observed data's variability can be attributed to the model, Q-square is a useful statistic. The SSE is the SSO multiplied by the SSO plus the SSE. If the Q-square number is near 1, the model fits the data well; if it is close to 0, the model does not match the data well. Table 9 indicates the results for predictive relevance where the best fit of the model is proven from $Q^2$ as it is greater than 0 as indicated in Table 9. The predicted relevance is confirmed, and the model can be used to obtain the same results in the future, which is critically related to the use of the model for improving the success of superhydrophobic coatings.

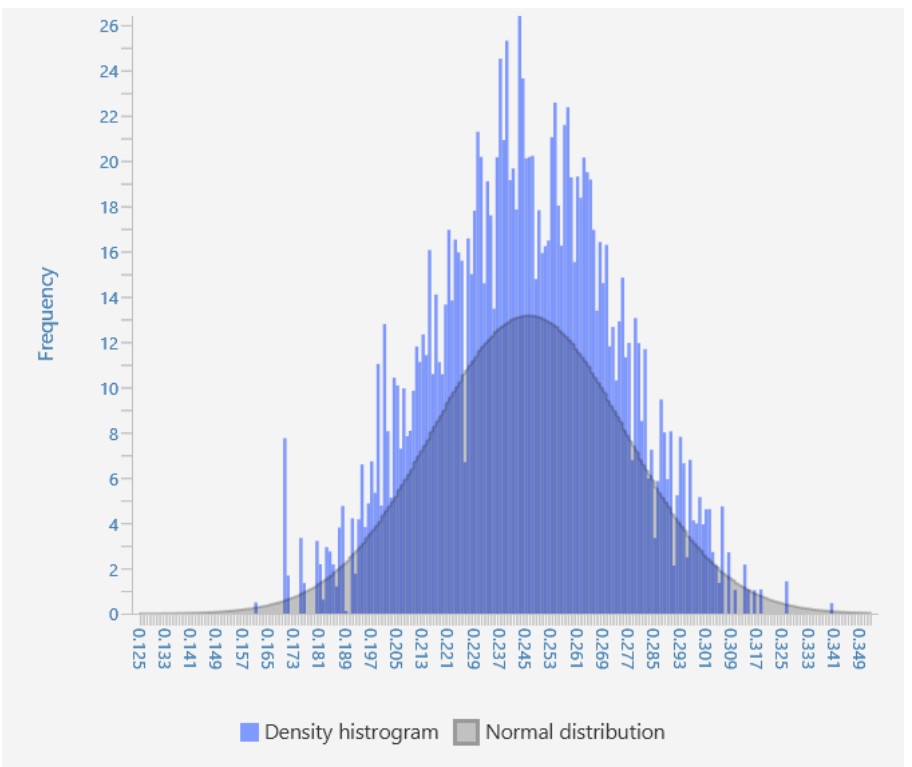

**Figure 7.** Path coefficient histogram for oil spill management construct.

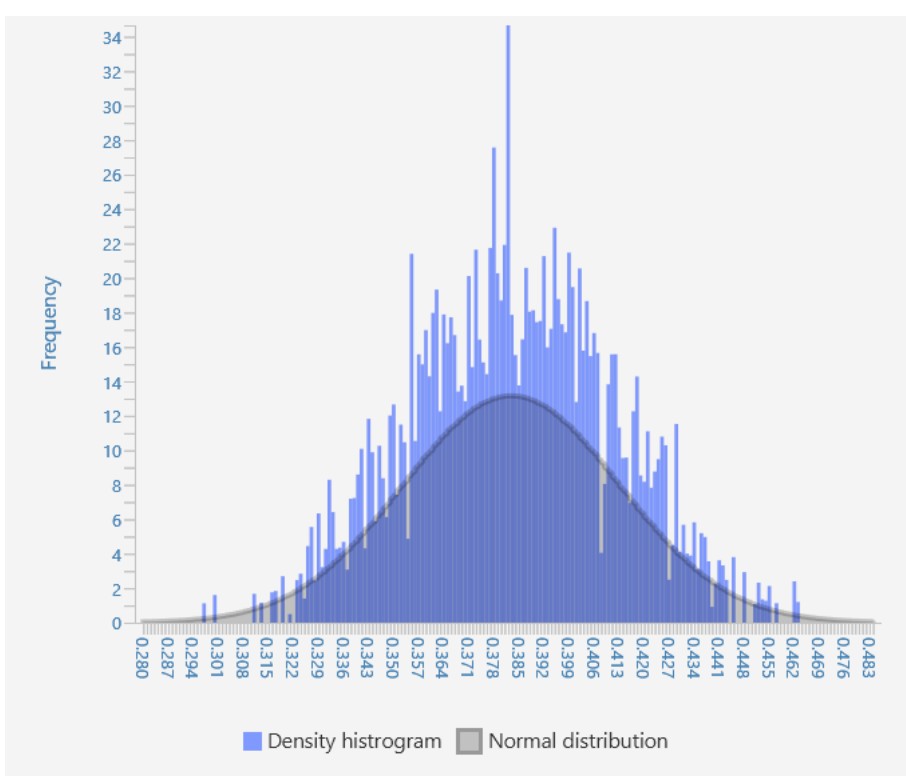

**Figure 8.** Path coefficient histogram for sustainability construct.

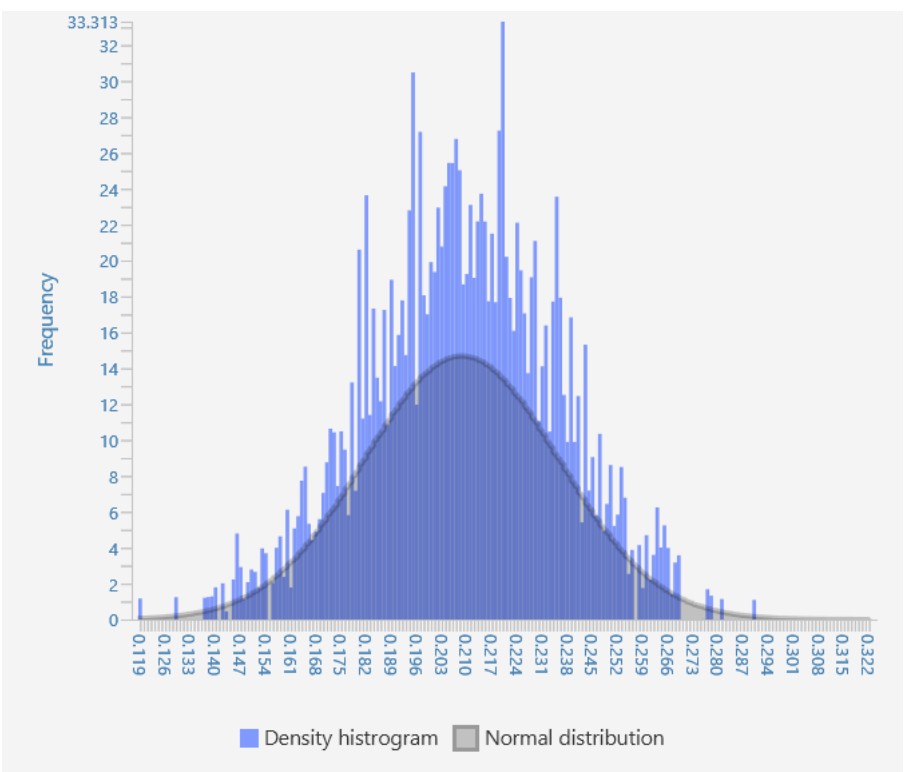

**Figure 9.** Path coefficients histogram of safety and economic construct.

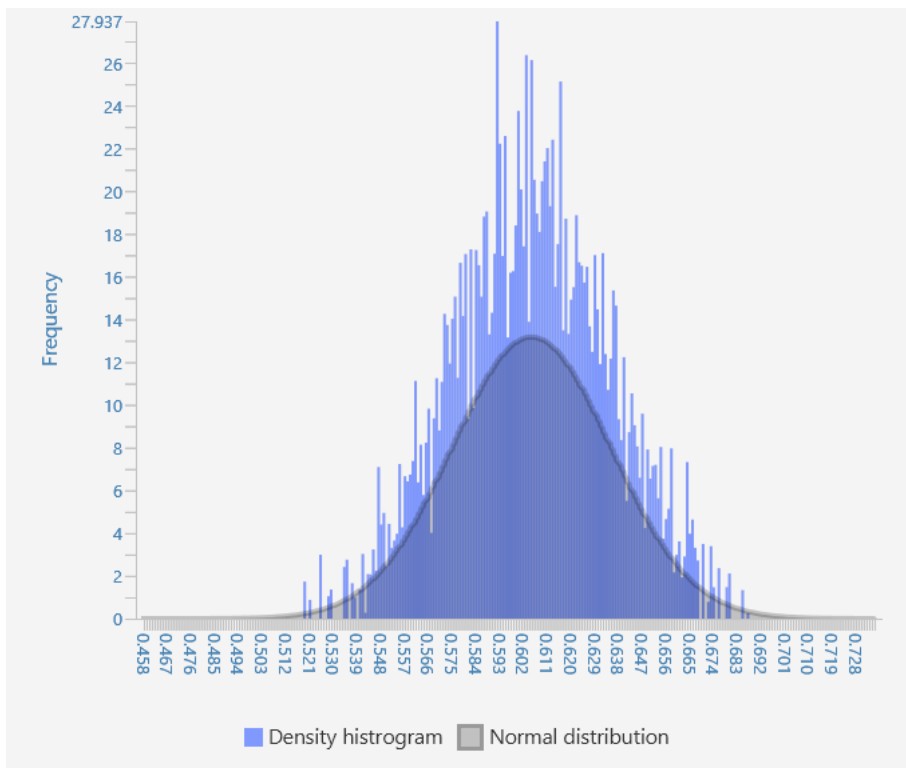

**Figure 10.** Path coefficients histogram of performance success construct.

**Table 9.** Predictive relevance results.

| Endogenous Latent Variable | SS0 | SSE | Predict-$Q^2$ |
|---|---|---|---|
| Success | 7075.000 | 5488.002 | 0.224 |

## 5. Discussion

The performance success construct includes S1 "Superhydrophobic coatings may enhance the effectiveness of oil processing & transportation systems, resulting in energy savings", S24 "Superhydrophobic coatings may make it simpler and quicker to clear up oil spills, hence reducing the time needed to resume regular operations", S14 "By decreasing the time necessary to clean up oil spills and the requirement of maintenance, superhydrophobic coatings may aid in increasing manufacturing productivity", S8 "By preventing oil from clinging to objects and preventing corrosion, superhydrophobic coatings may extend the life of oil and gas sector equipment", and S9 "By minimizing the requirement for maintenance and upkeep, superhydrophobic coatings may decrease the needed downtime for oil and gas sector equipment". The standard error observed in the performance success formative construct is 3.0% (SE = 0.03). For the items in this construct, the standard error was 8.4% for S1, 8.5% for S24, 9.5% for S14, 8.3% for S8, and 9% for S9. The error from the construct perspective needs to be less than 5%, and for individual items, it should be less than 10%. Acceptable error statistics are produced for the construct. The effectiveness of oil processing and transportation systems is expected to improve with superhydrophobic coatings, and that is the unique behavior observed from the analysis that is not found in Chen et al. (2021), Sarubbo et al. (2022), and Xu et al. (2021) [6,9,70]. From the existing research, it is evident that superhydrophobic coatings can improve the oil industry from a transportation perspective, but they do not provide other potential outcomes in the sense of extending the life of oil and gas equipment or reducing the necessary time to clean up oil spills, which is one of the most critical environmental disasters. Positive behavior is indicated where the ultimate impact of adopting superhydrophobic coatings will be better. Wong et al. (2021) argued the possible performance improvements in the oil and gas industry by increasing the implementation of superhydrophobic coatings [62]. This study indicated robust evidence of an increase in the performance of the oil and gas industry by implementing superhydrophobic coatings.

The sustainability success construct includes S2 "The resistance of superhydrophobic coatings to adverse state of the environment, including such high temperatures, humidity levels, and chemical exposure, makes them excellent for application in the oil and gas sector", S25 "By preventing oil from clinging to surface, superhydrophobic coatings may lessen the risk of fire and other dangers related to oil spills", S13 "Superhydrophobic coatings may minimize the quantity of oil that enters the environment after oil spills, hence minimizing the harm to species and ecosystems", and S4 "Companies in the oil and gas sector who utilize superhydrophobic coatings may gain a competitive edge by demonstrating their dedication to environmental preservation and operational efficiency". The standard error observed in the sustainability success formative construct is 3% (SE = 0.03). For the items in this construct, the standard error was 8.7% for S2, 9.0% for S25, 9.2% for S4, and 9.0% for S13. The error from the construct perspective needs to be less than 5%, and for items, it should be less than 10%. Acceptable error statistics are produced for the construct. The reduction in the risk of fire and other dangers is critically important as the main success factor of adopting superhydrophobic coatings, which alternately improve sustainability and also provide positive outcomes in the sense of improving the implementation of superhydrophobic coatings in the oil and gas industry. In accordance with Sebastian et al. (2021), Shagwira et al. (2020), and Yang et al. (2022), different behavior is observed in the case of the sustainability success construct, where the ultimate impact is critically visible from the model, and it is also the second most important construct in the model, ultimately improving the adoption of superhydrophobic coating in the oil and gas

industry [7,8,10]. The observed behavior further justifies the addition of information where the improvement in sustainability is critically important to the main success of choosing superhydrophobic coatings. Kiran et al. (2021) found the high significance of utilizing superhydrophobic coatings to produce sustainability in the oil and gas industry [35]. The findings are consistent with the existing trends, but the significance of items was different in this model.

The safety and economic success construct includes, S6 "These coatings may improve the effectiveness of oil processing as well as transport networks by lowering the quantity of oil that clings to surfaces", S19 "The application of superhydrophobic coatings may lower the expense of clearing up oil spills and repairing damaged infrastructure and equipment", and S15 "Superhydrophobic coatings may make surfaces simpler to clean, saving the time and effort required to keep equipment and infrastructure". The standard error observed in the safety and economic success formative construct is 2.7% (SE = 0.027). For the items in this construct, the standard error was 9.3% for S6, 9.1% for S19, and 8.6% for S15. The error from the construct perspective needs to be less than 5%, and for items, it should be less than 10%. Acceptable error statistics are produced for the construct. It was found that superhydrophobic coatings reduce the expense of oil clean-up during spill accidents and therefore justify the possible use of hydrophobic coatings in the oil and gas industry with critical success in the future. Furthermore, in accordance with Hajiyeva et al. (2021) and Reis et al. (2022), it is evaluated that the superhydrophobic coatings make the surfaces easy to clean, which ultimately provides an advantage to oil and gas industry equipment in extending their durability and service life [12,16]. The outcomes are entirely different from the existing research, as this study provides more significance in terms of maximizing safety and economic success when superhydrophobic coatings are used in the oil and gas industry. Peng et al. (2021) argued more positively for the safety of the oil and gas industry, as compared to the economic success aspect [39]. The findings of this study contributed by providing a clear view of the safety and economic aspect of the oil and gas industry.

The oil spill management construct includes S22 "Superhydrophobic coatings may help businesses comply with conservational requirements and standards by lowering the danger of oil spills as well as the environmental effect of spills" and S23 "Superhydrophobic coatings may boost the dependability of oil and gas sector equipment by avoiding oil from adherent to objects and causing corrosion". The standard error observed in the oil spill management formative construct is 3.0% (SE = 0.03). For the items in construct, the standard error was 9.6% for S22 and 8.7% for S23. The error from a construct perspective needs to be less than 5%, and for items, it should be less than 10%. Acceptable error statistics are produced for the construct. In accordance with Ren et al. (2021) and Zhang et al. (2021), it was found that superhydrophobic coatings can reduce the danger of oil spills, which is one of the significant advantages of oil spill management, and this uniquely identifies the possible success factor of superhydrophobic coatings in the oil and gas industry [29,30]. It has a significant impact on utilizing superhydrophobic coatings, as it can ultimately maximize oil spill management efficiency. According to Wei et al. (2019), a variety of applications exist specifically for oil spill disaster management with superhydrophobic coatings [21]. The study has shown expanded and practical success factors by indicating the specific ways through which oil spill management can be improved.

## 6. Conclusions

Structural equation modeling was used to assess how well superhydrophobic coatings had fared in the oil and gas construction sector. A correlation between the performance, management, sustainability, safety, and economic success of oil and gas construction projects and the use of superhydrophobic coatings was discovered. No statistically significant problems were found in the structural equation modeling findings. This study's results have important implications for building project managers in the oil and gas sector and for scientists studying the impact of superhydrophobic coatings on the effectiveness of such endeavors. The findings point to the potential importance of superhydrophobic coatings in

enhancing the efficiency, effectiveness, sustainability, safety, and profitability of oil and gas construction projects. This has the potential to enhance worker and community safety by decreasing the incidence of accidents. The study's findings show that superhydrophobic coatings may boost the profitability of oil and gas building projects by decreasing the time and money spent on maintenance and increasing the output per unit of effort. The research shows that using superhydrophobic coatings on oil and gas building projects helps to make them more environmentally friendly by cutting down on waste, increasing productivity, and decreasing the likelihood of corrosion. This has the potential to make businesses greener by decreasing their carbon impact and increasing the longevity of their operations. Because it demonstrates how structural equation modeling may be used to assess the performance of superhydrophobic coatings, this research adds to the existing body of information. Professionals and academics in the oil and gas construction business in Malaysia may utilize the findings of this study to direct future studies and influence industry-wide decisions.

## 7. Managerial and Empirical Implications

This research shows that using superhydrophobic coatings may greatly increase the success rate of oil and gas building projects. Managers may benefit from these data when deciding which coatings to use and how to apply them most effectively. The research may also be used to justify spending on superhydrophobic coatings, both in the lab and in the field. The research found that oil and gas construction projects benefited from the use of superhydrophobic coatings due to their ability to minimize corrosion, maintenance needs, and overall workload. Companies may benefit from these enhancements by reducing costs and increasing the likelihood of project success. The study's findings show that superhydrophobic coatings may aid project managers in minimizing corrosion risk, maximizing productivity, and minimizing liability. Management may benefit from this in that they will be able to better allocate resources, set priorities, and choose initiatives. The research found that by decreasing the potential for corrosion and increasing the durability of equipment and infrastructure, superhydrophobic coatings may increase the security of oil and gas development projects. This has the potential to improve the economic performance of projects, reduce costs, and boost revenue for businesses.

## 8. Limitations and Future Research

This research was conducted specifically for the Malaysian oil and gas construction sector; hence, its findings may not apply to other sectors or nations. We used self-reported data from experts in the field, which might include errors. Unfortunately, the complex interconnections that occur in real life may not be reflected in this research since it used structural equation modeling, which requires specific assumptions about the relationships between variables. No additional variables, such as climatic conditions or maintenance procedures, which may affect the performance of superhydrophobic coatings, were taken into account in the research. Superhydrophobic coatings have restrictions in the oil and gas construction business, despite their numerous potential advantages. Some of these drawbacks include poor durability, high cost, a difficult application method, limited compatibility with specific materials, and poor water repellency. Nevertheless, despite these drawbacks, superhydrophobic coatings can still be effective if the proper materials and application techniques are properly chosen and each project's unique requirements are taken into account. It is feasible to optimize the advantages of these coatings and reduce their disadvantages by taking these aspects into consideration, which will ultimately increase protection and performance in the oil and gas construction business. The research may not be indicative of the long-term efficacy of superhydrophobic coatings in the oil and gas construction sector since it only looks at a short time period. Lastly, it is possible that the sample size in this research is too small to make definitive conclusions.

**Author Contributions:** Conceptualization, A.W. and I.O.; methodology, A.W.; software, K.S.; validation, A.S.M.G.; formal analysis, A.W.; investigation, A.S.M.G., K.S.; resources, K.S., A.W.; data curation, K.S., A.S.M.G.; writing—original draft preparation, A.W., A.S.M.G.; writing—review and editing, K.S.; visualization, K.S.; supervision, I.O.; project administration, I.O.; funding acquisition, K.S. All authors have read and agreed to the published version of the manuscript.

**Funding:** This research received no external funding.

**Institutional Review Board Statement:** Not applicable.

**Informed Consent Statement:** Informed consent was obtained from all subjects involved in the study.

**Data Availability Statement:** The data is fully confidential and not available public access.

**Conflicts of Interest:** The authors declare no conflict of interest.

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
