# Peer review of "Evaluation of Success of Superhydrophobic Coatings in the Oil and Gas Construction Industry Using Structural Equation Modeling"

_coatings, doi:10.3390/coatings13030526_

Round 1
Reviewer 1 Report
1. Abstract should be specific, focusing more on findings of the research.
2. Introduction should be improved, some recent relevant research should be cited i.e. https://www.sciencedirect.com/science/article/pii/S1383586622021839
3. Quality of all figures should be improved significantly.
Author Response
|
S. No |
Reviewer Comments |
Action Taken |
Remarks |
|
1. |
Abstract should be specific, focusing more on findings of the research. |
Abstract adjusted, initial introduction shortened and findings part extended |
See Abstract |
|
2. |
Introduction should be improved, some recent relevant research should be cited i.e. https://www.sciencedirect.com/science/article/pii/S1383586622021839 |
Citation added in appropriate locations in introduction section of paper and overall introduction improved |
See Introduction |
|
3. |
Quality of all figures should be improved significantly. |
Resolution of all figures improved, realigned and adjusted in accordance with relevant content. |
See Figure 1 to Figure 10 |
Reviewer 2 Report
The manuscript "Evaluation of Success of Superhydrophobic Coatings in the Oil and Gas Construction Industry Using Structural Equation Modeling" has a well-developed analysis and model of the usefulness of superhydrophobic surfaces for the oil and gas industry (limited to Malaysia) presented over the past 3-5 years. However, literary analysis has the disadvantages of linking sentences by meaning. Therefore, the sentences in the paragraph on lines 151-158 link together, as well as with the previous text. Also there are other places with unrelated common thought...Also correct the texts with smaller font (lines 217 etc.). In the title of table 1, remove the period.
At the same time, for the reader of this work, it is of high utility to characterize not only usefulness, but also disadvantages for SHA coatings. Probably, one of the main drawbacks is the difficulty of creating such surfaces in industrial volumes. One can trace the trend that with a higher level of protective properties of the superhydrophobic surface, the complexity and staging of their production technology increases. Also, the high stability of the SHA properties and the reproduction of a similar quality of the layers are also important. The introduction of the authors should be better expanded with this information, which is useful to the reader. In this regard, according to the analyzed articles in this paper, it is also useful to make a connection between the quality of SHA surfaces and their shortcomings.
Author Response
|
S. No |
Reviewer Comments |
Action Taken |
Remarks |
|
1. |
The manuscript "Evaluation of Success of Superhydrophobic Coatings in the Oil and Gas Construction Industry Using Structural Equation Modeling" has a well-developed analysis and model of the usefulness of superhydrophobic surfaces for the oil and gas industry (limited to Malaysia) presented over the past 3-5 years. |
- |
The authors wish to thank the reviewer for his valuable time and kind effort. By addressing the comments, the quality of the manuscript has been enhanced. |
|
2. |
However, literary analysis has the disadvantages of linking sentences by meaning. Therefore, the sentences in the paragraph on lines 151-158 link together, as well as with the previous text. Also there are other places with unrelated common thought...Also correct the texts with smaller font (lines 217 etc.). In the title of table 1, remove the period. |
All the sentences having no link or connection with meaning improved and citations adjusted. Sentences in the paragraph on lines 151 to 158 were entirely removed as they were inconsistent with the flow of content. In introduction, related studies and methodology, unrelated common sentences were deleted to reshape the content. All the font size on lines including 217, corrected and matched with paragraph style. Period in the title of table 1 was removed. |
See lines 151 to 158 See Section, Introduction, Related Studies and Methodology See line 217 See Title of Table 1 |
|
3. |
At the same time, for the reader of this work, it is of high utility to characterize not only usefulness, but also disadvantages for SHA coatings. Probably, one of the main drawbacks is the difficulty of creating such surfaces in industrial volumes. One can trace the trend that with a higher level of protective properties of the superhydrophobic surface, the complexity and staging of their production technology increases. Also, the high stability of the SHA properties and the reproduction of a similar quality of the layers are also important. The introduction of the authors should be better expanded with this information, which is useful to the reader. In this regard, according to the analyzed articles in this paper, it is also useful to make a connection between the quality of SHA surfaces and their shortcomings |
Disadvantages are discussed in literature review from the latest and relevant literature. The authors introduction expanded in accordance with the citation and the relevant tax. The connection between the qualities of SHA coatings and shortcomings were discussed in literature review and limitations section of the paper |
See Line 85 to 171 See Line 34 to 84 |
Reviewer 3 Report
Comments to the Authors:
The authors of this paper present an interesting evaluation of the success factors connected with the adoption of superhydrophobic coatings in the oil and gas construction industry of Malaysia. It is a good work, nevertheless, some details should be considered by the authors:
GENERAL COMMENT: It is a well written work. However, although a large amount of data and results are presented in this article, the discussion about possible errors (or quantification of errors) is limited. More comments could be added about a possible evaluation of errors.
Nevertheless, the results support the authors’ conclusion. Thus, I think that this paper may be published.
Author Response
|
S. No |
Reviewer Comments |
Action Taken |
Remarks |
|
1. |
The authors of this paper present an interesting evaluation of the success factors connected with the adoption of superhydrophobic coatings in the oil and gas construction industry of Malaysia. It is a good work, nevertheless, some details should be considered by the authors: |
- |
The authors wish to thank the reviewer for his valuable time and kind effort. By addressing the comments, the quality of the manuscript has been enhanced. |
|
2. |
It is a well written work. However, although a large amount of data and results are presented in this article, the discussion about possible errors (or quantification of errors) is limited. More comments could be added about a possible evaluation of errors.
Nevertheless, the results support the authors’ conclusion. Thus, I think that this paper may be published. |
Standard errors for the constructs are discussed. Standard errors for each of the item are discussed. |
In Discussion Section,
See Line 495-499 Line 520-524 Line 543-547 Line 563-567 |
Reviewer 4 Report
The manuscript "coatings-2231917" by Lastname et al. reported Evaluation of Success of Superhydrophobic Coatings in the Oil and Gas Construction Industry Using Structural Equation Modeling. After review, this study currently unpublishable. The authors have to make major changes. The authors should refer to the following comments to improve their work:
a. The paper does not follow a continuous structure, especially from the introduction to the beginning of the results (pages 1 to 9). Many sentences are repetitive and explain the same purpose but in different words. For example, it is enough to explain the importance of your study at the end of the introduction, it is not necessary to repeat it in different parts. I suggest that you summarize these contents (pages 1 to 9).
b. Please improve the quality of the Figures.
c. The titles of the horizontal and vertical axes as well as the numbers in Figures 4, 5, and 7 cannot be read.
d. Please improve the discussion.
e. The language of the manuscript should be checked.
Author Response
Please see the attachment.
|
S. No |
Reviewer Comments |
Action Taken |
Remarks |
|
|
The manuscript "coatings-2231917" reported Evaluation of Success of Superhydrophobic Coatings in the Oil and Gas Construction Industry Using Structural Equation Modeling. After review, this study currently unpublishable. The authors have to make major changes. The authors should refer to the following comments to improve their work. |
- |
The authors wish to thank the reviewer for his valuable time and kind effort. By addressing the comments, the quality of the manuscript has been enhanced. |
|
1. |
The paper does not follow a continuous structure, especially from the introduction to the beginning of the results (pages 1 to 9). Many sentences are repetitive and explain the same purpose but in different words. For example, it is enough to explain the importance of your study at the end of the introduction, it is not necessary to repeat it in different parts. I suggest that you summarize these contents (pages 1 to 9). |
From page 1 to 9, any extra and repetition is removed. Introduction, related theory and methodology sections adjusted by deleting the parts where importance of study was repeated. Content is now summarized with best possible structure, avoiding repetition.
|
See Page 1 to 9 |
|
2. |
Please improve the quality of the Figures. |
All figures are improved for best possible view.
|
See Figure 1 to 10 |
|
3. |
The titles of the horizontal and vertical axes as well as the numbers in Figures 4, 5, and 7 cannot be read. |
All figures replaced with enhanced resolution and now axes and titles are clearly visible.
|
See Figure 1 to 10 |
|
4. |
Please improve the discussion. |
Discussion section is improved by adding latest literature in all constructs. |
See Discussion Section |
|
5. |
The language of the manuscript should be checked. |
Grammarly is used to refine the language and optimize according to requirements of paper. |
- |
Round 2
Reviewer 4 Report
Accept in present form.